# Transmission of *Escherichia coli* Causing Pyometra between Two Female Dogs

**DOI:** 10.3390/microorganisms10122465

**Published:** 2022-12-14

**Authors:** Rafael Gariglio Clark Xavier, Clarissa Helena Santana, Paloma Helena Sanches da Silva, Flávia Figueira Aburjaile, Felipe Luiz Pereira, Henrique César Pereira Figueiredo, Patrícia Maria Coletto Freitas, Renato Lima Santos, Rodrigo Otávio Silveira Silva

**Affiliations:** Veterinary School, Federal University of Minas Gerais, Antônio Carlos Avenue 6627, Belo Horizonte 31270-090, Brazil

**Keywords:** ExPEC, EnPEC, UPEC

## Abstract

Despite its clinical relevance, the pathogenesis of canine pyometra remains poorly understood. To date, it is recognized as a non-transmissible infectious disease. In this study, the simultaneous occurrence of pyometra and *Escherichia coli* in two cohabitant female dogs underwent in-depth investigation due to the hypothesis of transmission between these animals. Two 5-year-old Chow Chow dogs (namely, dogs 23 and 24—D23 and D24) were referred to a veterinary hospital with suspected pyometra. Both animals showed prostration, anorexia, and purulent vulvar discharge over a 1-week period. After ovariohysterectomy, uterine tissue, uterine contents, and rectal swabs were collected for histopathological and microbiological analysis. Uterine histology demonstrated purulent material and multifocal necrosis with endometrial ulceration, and a morphological diagnosis of pyometra was confirmed. Furthermore, *E. coli* from the same phylogroup (B2) and positive for the same virulence factors with the same antimicrobial susceptibility profile was isolated from the uterine contents of both dogs and the rectum of D23. Conversely, the *E. coli* strains recovered from D24 differed in phylogroup (one isolate), virulence factors (all three isolates), and antimicrobial susceptibility (all three isolates). Enterobacterial repetitive intergenic consensus polymerase chain reaction (ERIC-PCR) suggested that all isolates from the uterine content of both dogs and the rectal swab of D23 were 100% the same, but different from all isolates in the rectal swab of D24. One isolate from the uterine content of each animal as well as rectal swabs were subjected to whole-genome sequencing (WGS). Both whole-genome multilocus sequence typing(wgMLST) and single-nucleotide polymorphism (SNP) analysis supported the hypothesis that the isolates from the uterine content of both animals and the rectal swab of D23 were clonal. Taken together, these clinical features, pathology, microbiology, and molecular findings suggest, to the best of our knowledge, the first transmission of *E. coli* associated with pyometra between two animals. These results could impact the management of sites where several females cohabit in the same local area such as kennels.

## Short Communication/Note

Pyometra is the most frequently observed reproductive disease in bitches, affecting up to 25% of unspayed females [1,2]. The disease is characterized by bacterial infection of the uterus with local and potentially fatal systemic clinical manifestations such as prostration, anorexia, purulent vulvar discharge, sepsis, and multi-organ dysfunction [3,4,5]. Despite its relevance, the pathogenesis of this disease remains poorly understood. It is believed that bacterial species may cause pyometra to ascend from the host’s intestinal tract, causing a non-transmissible opportunistic infection [2,5,6]. In this study, we describe an in-depth investigation of the possible transmission of *Escherichia coli* associated with pyometra in two bitches.

A 5-year-old female Chow Chow (D23) was referred to the Veterinary Hospital of the Universidade Federal de Minas Gerais (VH-UFMG) with a purulent vulvar discharge. In addition to the vulvar discharge, the animal was hyperthermic (41 °C) and showed signs of prostration, anorexia, and diarrhea, suggestive of an open pyometra. The examinations also revealed anemia (hematocrit: 36%; RV: 37–55%), thrombocytopenia (platelets: 124,000/mm^2^; RV: 175,000–500,000/mm^2^, and azotemia (creatinine: 1.57 mg/dL; RV: 0.5–1.5 mg/dL). The animal underwent ovariohysterectomy (OHE) surgery. Just after the procedure, a sample from the uterine content and feces from the rectal ampulla of the dog were collected by needle aspiration and swab, respectively. The samples were refrigerated at 4 °C until processing for a maximum of 24 h. Samples from the uteri and ovaries were collected for histopathological analysis.

After 5 days, another female dog (D24) from the same litter and cohabiting with D23 was referred to VH-UFMG with similar symptoms including prostration, anorexia, and purulent vulvar discharge. These two dogs were the only animals in their household. Examination results indicated anemia (erythrocytes: 4.48 million/mm^2^; RV: 5.5–8.5 million/mm^2^ and hematocrit 25%), leukocytosis (leukocytes: 28,200 mm^2^; RV: 6000–17,000 mm^2^), thrombocytopenia (platelets: 90,000 mm^2^), decreased blood urea nitrogen (BUN: 17.69 mg/dL; RV: 20–56 mg/dL), increased alkaline phosphatase (ALP: 157 U/L; RV: 40–156 U/L). This animal also underwent OHE surgery and again, uterine tissue, uterine content, and rectal swab samples were collected. After surgery, both animals were treated with amoxicillin/clavulanic acid and metronidazole.

Samples from the uteri and ovaries of both animals were fixed by immersion in 10% buffered formalin for 24 h, processed for paraffin embedding, and sectioned (3-µm thick), and stained with hematoxylin and eosin for histopathology. The uteri of both bitches were enlarged and filled with a significant amount of purulent brown material. Both animals were in diestrus, which was confirmed by the discovery of multiple corpora lutea. In addition, the ovaries of one bitch (D23) had neutrophilic arteritis and fibrinous thrombi, partially occluding the artery.

Microscopically, in both animals, the uterine lumen was filled with many neutrophils, fibrin, bacterial aggregates, and, in D23, also blood. There was a severe diffuse neutrophilic and lympho-histioplasmacytic endometrial inflammatory infiltrate, with marked neutrophilic exocytosis into the uterine lumen and endometrial glands. In D23, there was severe multifocal necrosis with endometrial ulceration extending to the superficial endometrial glands, with intense endometrial hemorrhage, fibrin deposition, and moderate fibroplasia (Figure 1); many other endometrial veins were filled with fibrin thrombi, which partially occluded the lumen. In D24, there was mild multifocal endometrial ulceration. These concurrent findings are highly indicative of the histopathological lesions observed in cases with severe pyometra [7,8]. Endometrial necrosis and ulceration may also be observed in pyometra cases; however, which determines the intensity of necrosis in each case is not well-defined [8]. In D23, the remaining endometrial epithelial cells and in D24, epithelial cells of the luminal endometrial epithelium and superficial endometrial glands were columnar with finely vacuolated cytoplasm, morphologically indicative of decidual reaction. In both animals, endometrial glands were diffusely and markedly ecstatic with the accumulation of neutrophils and mucous [8] (Figure 1). In addition, a multifocal moderate histioplasmacytic and neutrophilic inflammatory infiltrate extended into the myometrium. Therefore, a morphological diagnosis of pseudoplacentational endometrial hyperplasia and pyometra was established in both bitches [7]. Interestingly, as observed in these bitches, a recent study described the high frequency of the association of pseudoplacentational endometrial hyperplasia with pyometra in female dogs [9].

The uterine contents and rectal swabs were plated on Mueller–Hinton agar (Sparks, BD, USA) supplemented with 5% equine blood and in MacConkey agar (Kasvi, São José dos Pinhais, Brazil), followed by aerobic and anaerobic incubation at 37 °C for 48 h. Twelve lactose-fermenting colonies from each sample (48 isolates in total) were subjected to a species-specific polymerase chain reaction (PCR) to identify *E. coli* [10]. Thus, the isolates were subjected to PCR to determine *E. coli* phylogroups [11] and to detect the main virulence genes associated with the extraintestinal pathogenic *E. coli* (ExPEC) pathotype, namely, fimbriae type I (*fimH*), fimbriae type I central region (*focG*), fimbriae type P (*papC* and *papG* alleles II and III), fimbriae type S (*sfaS*), cytotoxic necrotizing factor type 1 (*cnf1*), α-hemolysin (*hlyA*), uropathogenic specific protein (*usp*), aerobactin (*iutA*), and serum resistance (*traT*) [12,13]. Additionally, antimicrobial susceptibility to amoxicillin/clavulanic acid, ampicillin, ceftiofur, ciprofloxacin, enrofloxacin, neomycin, gentamicin, doxycycline, oxytetracycline, and trimethoprim/sulfamethoxazole was assessed using the disk diffusion method and interpreted according to the Clinical and Laboratory Standards Institute [14,15].

All *E. coli* strains isolated from the uterine contents of both female dogs were classified into the same phylogroup (B2) (Figure 2). This result was not surprising as *E. coli* is the most common bacterial organism found in a pyometra, and these isolates tend to cluster mainly in phylogroup B2 [5,16,17]. Furthermore, the uterine *E. coli* isolates also had the same ExPEC virulence factor-encoding genes (*fimH*, *focG*, *papC*, *papG*, *cnf1*, *hlyA*, and *usp*) (Figure 2), virulence factors also commonly associated with endometrial pathogenic *E. coli* [16,18]. These virulence factors, mainly the fimbriae-encoding genes (*fimH*, *focG*, *papC* and *papG*), are of great relevance for the establishment of *E. coli* infections in the canine uterus [13,16,19]. Additionally, *E. coli* strains from the uterine contents had no antimicrobial resistance (Figure 2).

Interestingly, the *E. coli* isolates recovered from the D23 rectal swab showed the same phylogroup, virulence factors, and antimicrobial profile as the isolates recovered from the uterus of D23 and D24. In contrast, the strains isolated from the D24 rectal swab differed in phylogroup (four isolates), virulence factors (all twelve isolates), and antimicrobial susceptibility (all twelve isolates). Based on these results, two hypotheses were raised: first, that the *E. coli* strain colonizing D23′s rectum ascended to the uterus of this animal, causing the infection; second, that this *E. coli* strain was transmitted to D24, causing pyometra.

To further investigate the possible clonal origin of the isolates, six *E. coli* strains from each dog (tree per isolation) were fingerprinted by Enterobacterial repetitive intergenic consensus polymerase chain reaction (ERIC-PCR) [20,21] and random repetitive extragenic palindromic (REP)-PCR [22,23]. These techniques were successfully used in previous investigations of outbreaks caused by Enterobacteriaceae [24]. Furthermore, both ERIC-PCR and REP-PCR suggested that all isolates from the uterine content of both dogs and the rectal swab of D23 were 100% similar, but different from all isolates from the rectal swab of D24 (similarity ≤84.6%) (Figure 2), reinforcing the suspicion that *E. coli* was transmitted from D23 to D24, causing pyometra.

To better evaluate the hypothesis of transmission of *E. coli* associated with pyometra between the two bitches, four strains were subjected to whole-genome sequencing, currently considered the technique with the highest accuracy and resolution for these cases [25]. One isolate from each site (uterus and rectum) from each animal was included (marked in red in Figure 2). Genomic deoxyribonucleic acid was extracted using the Maxwell 16^®^ Research Instrument (Promega, Madison, WI, USA) combined with lysozyme (10 mg/mL) and proteinase K (20 mg/mL). The genome was sequenced using the Ion Torrent PGMTM in a mate-pair sequencing kit with an insert size of 3 kbp (~144-fold coverage) and with a fragment sequencing 400 bp kit (~318-fold coverage). The quality of the raw data was analyzed using FastQC (Babraham Bioinformatics), trimming of reads to remove adapters and 3′ ends with Phred’s quality score <20 was conducted with an in-house-script (available at https://github.com/aquacen/fast_sample), and assembly was performed using SPAdes 3.5.0 [26]. With default parameters, automatic annotation was performed using Prokka 1.10 (Rapid Bacterial Genome Annotation) software [27]. VirulenceFinder 2.0, ResFinder 4.1, and SerotypeFinder 2.0 [28,29,30,31,32,33] were used to identify virulence factors, antimicrobial resistance genes, and mutations, and to predict the O serotype. Multilocus sequence typing (MLST) 2.0 was used to determine sequencing types (ST) according to the Achtman MLST scheme [28,34,35,36] and Phyloviz v 2.0, using the goeBURST algorithm [37,38], was used to infer the population structure with clonal complexes (CCs) composed of all strains sharing at least six identical alleles (single-locus variant). Whole-genome MLST of the four isolates was performed using Ridom SeqSphere+ 4.1.9 [39] and 13 reference *E. coli* strains from previous studies on humans were included for comparison purposes. These strains were subjected to single-nucleotide polymorphism (SNP) analysis using CLC Workbench software v 6 (Qiagen, Aarhus, Denmark). Parameters for alignment were settled as mismatch cost = 2; insertion/deletion cost = 1. Parameters to SNP calls were defined to minimum coverage = 10; minimum variant frequency = 50%; filter 454/ion homopolymer indels = 1. Other parameters for alignment and SNP calls remained as the default.

The three likely clonal isolates were classified into a new sequence type on the same CC of ST372 (Figure 3; Table 1), uropathogenic *E. coli* previously described in several reports on dogs and humans, commonly causing genitourinary infection [40,41,42,43,44,45]. Furthermore, the SNP analysis confirmed the clonal relationship between the isolates from the uterine content of D23 and D24 and the strains isolated from the rectum of D23, with a maximum difference of 22 SNPs. 

Moreover, as expected, the isolate from the rectum of D24 differed greatly from these three isolates (Figure 4). Together, these results confirm that the strain colonizing the rectum of D23 ascended to the uterus of this bitch, causing pyometra. This is the currently accepted pathogenesis of this disease [2]. On the other hand, this same bacterium was later transmitted to the uterus of D24, causing pyometra. This is the first report of a likely transmission of *E. coli* pyometra between two female dogs.

Despite its clinical relevance, the pathogenesis of canine pyometra remains poorly understood. So far, pyometra has been recognized as a non-transmissible infectious disease [1,2,46]. The present study describes a likely transmission of *E. coli* pyometra between two dogs. This hypothesis was first raised after two cohabiting bitches from the same litter developed pyometra in a short period. The genetic similarity seen in the genome of these three strains confirmed the clonal origin, reinforcing the hypothesis of transmission. Although it is impossible to determine the source of contamination for D24, the source was direct or indirect contact with feces or vulvar discharge from D23. Previous studies have suggested that, in addition to feces, uterine contents may be a source of dissemination of pathogenic *E. coli* to the environment, possibly contaminating other hosts [13,19,47]. It is worth mentioning that D23 had an open pyometra, and according to the owner, a constant purulent discharge had been observed.

A noteworthy point is that the isolate causing pyometra in both dogs was classified in the same CC of ST372, a well-known pathogen for dogs and humans. In addition, several studies with ExPEC isolated from dogs have shown a high similarity of these isolates with those recovered from humans cohabiting in the same environment [48,49,50,51].

The clone that caused pyometra in both dogs was also isolated from the gut microbiota of dog 23 (D23), but not from the gut of dog 24 (D24). However, it is not possible to completely rule out that this clone was present in the gut of D24. Furthermore, it is also possible that both dogs acquired this clone from an unknown common source. Importantly, these two were the only dogs sharing their environment. Finally, it is also important to consider that some breeds seem to be at increased risk of developing pyometra including Chow Chow [52,53,54].

For the first time, this study describes the possible transmission of *E. coli* pyometra in dogs. Our findings suggest that in sites with more than one unspayed female, animals with suspected or confirmed pyometra should be isolated from other bitches until clinical resolution. This finding will be relevant to kennel managers, owners with several dogs, and even hospitals.

## Figures and Tables

**Figure 1 microorganisms-10-02465-f001:**
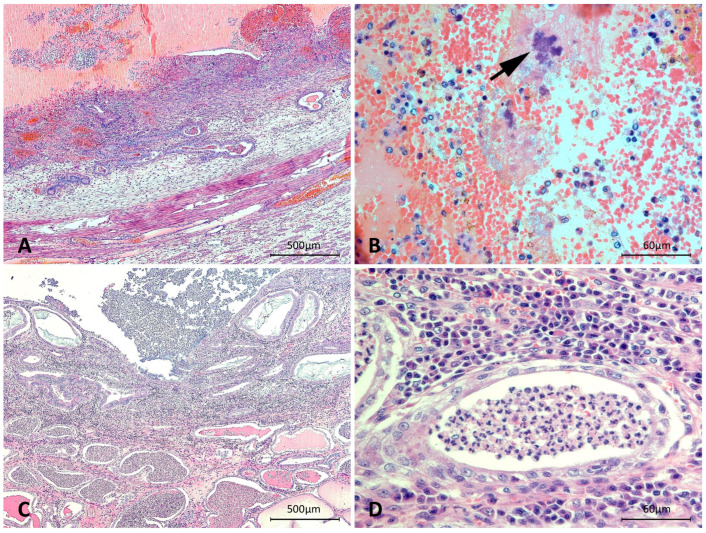
Canine uteri. (**A**) D23—uterine lumen filled with large amounts of blood and increased cellularity in the endometrium, with severe necrosis, endometrial ulceration, and hemorrhage. Endometrial glands are markedly ecstatic. (**B**) D23—higher magnification of A: hemorrhagic uterine luminal content, with neutrophils, plasma cells, fibrin, and bacterial aggregates (arrow). (**C**) D24—severe diffuse endometritis, with endometrial glands markedly ecstatic with an accumulation of inflammatory cells and mucous, and superficial luminal and glandular epithelium composed of columnar and finely vacuolated cells. (**D**) D24—higher magnification of C: severe diffuse interstitial lympho-histioplasmacytic infiltrate, and an endometrial gland filled with neutrophils. Hematoxylin and eosin, scale bars = 500 µm (**A**,**C**), 60 µm (**B**,**D**).

**Figure 2 microorganisms-10-02465-f002:**
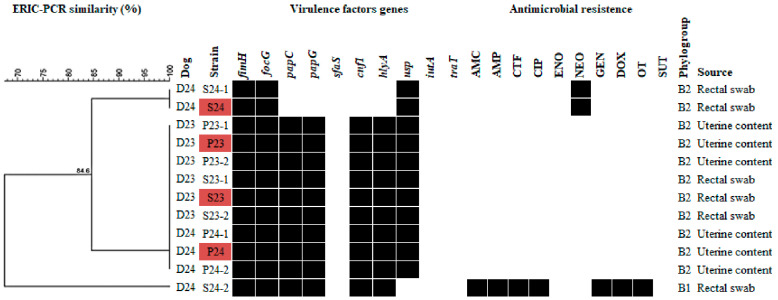
Enterobacterial repetitive intergenic consensus polymerase chain reaction (ERIC-PCR) similarity, virulence factors, phylogroup, and antimicrobial profile of *E. coli* isolated from rectal swabs and uterine contents of two cohabiting bitches (D23 and D24). Legend: P: uterine content; S: rectal swab, AMC: amoxicillin/clavulanic acid, AMP, ampicillin, CTF: ceftiofur, CIP: ciprofloxacin, ENO: enrofloxacin, NEO: neomycin, GEN: gentamicin, DOX: doxycycline, OT: oxytetracycline, and SUT: trimethoprim/sulfamethoxazole. Isolates marked in red were selected for whole genome sequencing.

**Figure 3 microorganisms-10-02465-f003:**
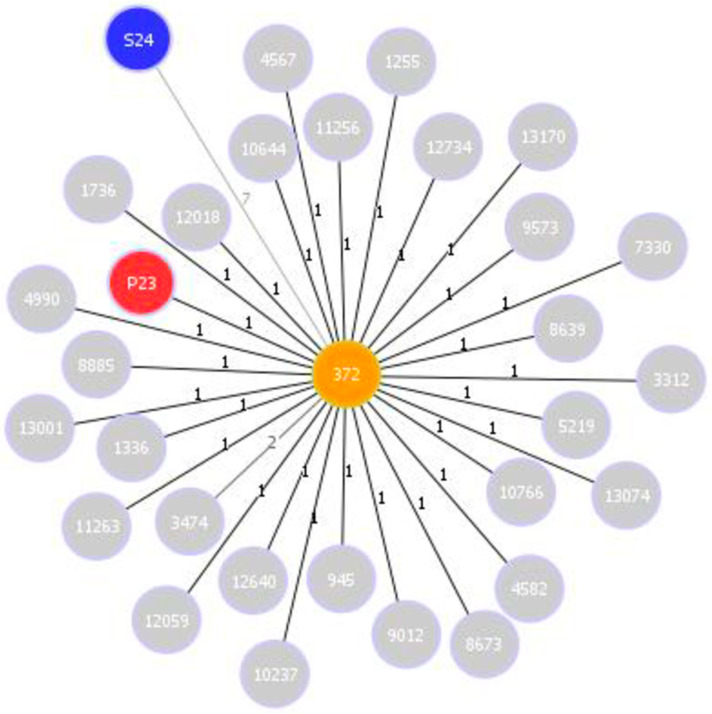
goeBURST population snapshot of clonal complex (CC) 372. Only isolates with a single locus variance (SLV) and isolates from the present study were included: the S24 isolate is marked in blue; ST372 (founder) is marked in orange, P23 is marked in red, and other isolates obtained from the public *E. coli* MLST database are marked in gray; line numbers indicate allelic variance.

**Figure 4 microorganisms-10-02465-f004:**
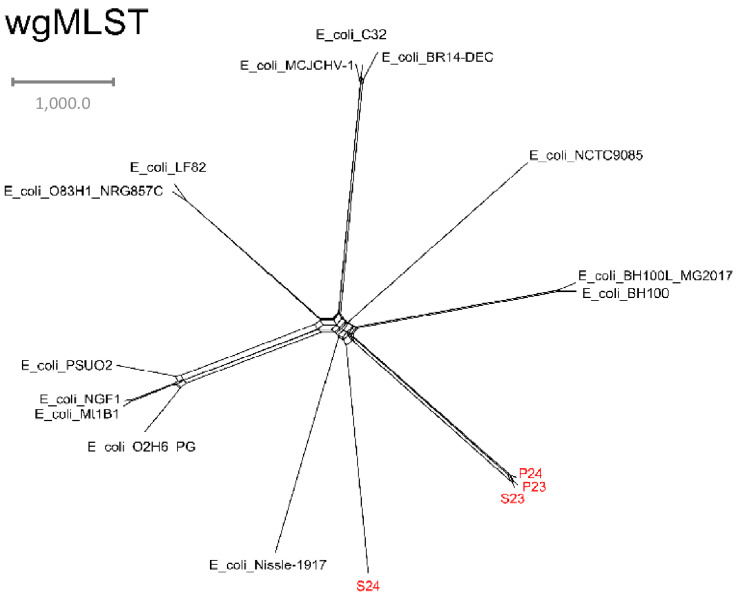
wgMLST phylogeny tree including the present study isolates (in red) and reference strains added for comparison purposes.

**Table 1 microorganisms-10-02465-t001:** Results of virulence factors, resistance gene detection, and multilocus sequence typing (MLST) of the four *E. coli* isolates recovered from the uterine contents and rectal swabs of two cohabiting Chow Chow bitches.

Animal	Source	Sample ID	Accession Number	O Serotype ^1^	Antimicrobial Resistance Genes	Virulence Factors
D23	Uterine content	P23	JAMJIL000000000	O4:H31	*mdf(A)* *sitABCD*	*papC*, *cnf1*, *focI*, *hra*, *papA_F13*, *focG*, *usp*, *chuA*, *cea*, *clbB*, *focCsfaE*, *fyuA*, *ibeA*, *iroN*, *irp2*, *iss*, *mchB*, *mchC*, *mchF*, *mcmA*, *ompT*, *sitA*, *terC*, *vat,* and *yfcV*
	Rectal swab	S23	JAMJIK000000000			
D24	Uterine content	P24	JAMJIJ000000000			
	Rectal swab	S24	JAMJII000000000	O5:H22	*mdf(A)* *sitABCD*	*gad*, *kpsE*, *kpsMII*, *pic*, *sfaD*, *tcpC*, *focG*, *usp*, *chuA*, *cea*, *clbB*, *focCsfaE*, *fyuA*, *ibeA*, *iroN*, *irp2*, *iss*, *mchB*, *mchC*, *mchF*, *mcmA*, *ompT*, *sitA*, *terC*, *vat,* and *yfcV*

^1^ Predicted serotype by SerotypeFinder 2.0 [29,33].

## Data Availability

The whole-genome shotgun project was deposited in GenBank/NCBI under the BioProject accession number PRJNA824036, Biosample SAMN27382112 (P23), SAMN27382113 (S23), SAMN27382114 (P24), and SAMN27382115 (S24). Genome accession numbers: JAMJIL000000000 (P23), JAMJIK000000000 (S23), JAMJIJ000000000 (P24), and JAMJII000000000 (S24) [https://www.ncbi.nlm.nih.gov/bioproject/PRJNA824036/, accessed on 8 September 2022].

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
