# Peer review of "Transmission of Escherichia coli Causing Pyometra between Two Female Dogs"

_microorganisms, 2022, doi:10.3390/microorganisms10122465_

Round 1

Reviewer 1 Report

General comments: This is an interesting paper describing two littermate females having a pyometra at the same time with the same strains of E. coli in the uterus. The pyometra is not transmissible but the bacteria are. Cystic endometrial hyperplasia or pseudoplacentational endometrial hyperplasia precedes a pyometra. Throughout the entire document I have an issue with saying that the pyometra was transmitted. The bacteria were transmitted. A pyometra is not infectious but the bacteria are. Both dogs were from the same litter, raising also the possibility of a similar immune system and similar predisposition for developing a pyometra. . Most publications abbreviate ovariohysterectomy with OHE rather than OH.

Line 27: 100% similar should be replaced with 100% the same (items can’t be 100% similar).

Line 90: Pseudoplacentational is misspelled.

Line 120-121: I wouldn’t say E. coli is the leading cause; I would phrase this differently. Perhaps: This result was not surprising as E. coli is the most common bacterial organism found in a pyometra. The actual cause of a pyometra is still poorly understood.

Author Response

Reviewer 1

This is an interesting paper describing two littermate females having a pyometra at the same time with the same strains of E. coli in the uterus. The pyometra is not transmissible but the bacteria are. Cystic endometrial hyperplasia or pseudoplacentational endometrial hyperplasia precedes a pyometra. Throughout the entire document I have an issue with saying that the pyometra was transmitted. The bacteria were transmitted. A pyometra is not infectious but the bacteria are. Both dogs were from the same litter, raising also the possibility of a similar immune system and similar predisposition for developing a pyometra. Most publications abbreviate ovariohysterectomy with OHE rather than OH.

Authors: Thanks for this comment. As suggested, we have modified the manuscript to emphasize that pyometra-associated bacteria were transmitted, not the pyometra itself. We also abbreviated ovariohysterectomy to OHE instead of OH.

Line 27: 100% similar should be replaced with 100% the same (items can’t be 100% similar).

Authors: Revised as suggested.

Line 90: Pseudoplacentational is misspelled.

Authors: Revised as suggested.

Line 120-121: I wouldn’t say E. coli is the leading cause; I would phrase this differently. Perhaps: This result was not surprising as E. coli is the most common bacterial organism found in a pyometra. The actual cause of a pyometra is still poorly understood.

Authors: Revised as suggested.

Reviewer 2 Report

The Authors reported cases of pyometra that occurred in two bitches belonging same litter and were caused by Escherichia coli. The microorganism isolated belonged to the same phylogroup (B2) and was positive for the same virulence factors with the same antimicrobial susceptibility profile was isolated from the uterine contents of both dogs and the rectum.

The authors hypothesize ascendent infection from the rectum and the possible transmission between dogs.

Although the report may be interesting, the results and the scientific soundness of the paper are very limited and are based on exclusively the hypothesis regarding the transmission.

Based on the below report, in my opinion, the results do not have quality enough to continue the process of publication. 

Author Response

The Authors reported cases of pyometra that occurred in two bitches belonging same litter and were caused by Escherichia coli. The microorganism isolated belonged to the same phylogroup (B2) and was positive for the same virulence factors with the same antimicrobial susceptibility profile was isolated from the uterine contents of both dogs and the rectum. The authors hypothesize ascendent infection from the rectum and the possible transmission between dogs. Although the report may be interesting, the results and the scientific soundness of the paper are very limited and are based on exclusively the hypothesis regarding the transmission. Based on the below report, in my opinion, the results do not have quality enough to continue the process of publication.

Authors: In the present manuscript, we describe in details the clinical features, pathology, microbiology, and molecular findings of two cases of pyometra. Taking together, all findings, including the SNPs analysis, strongly suggest the first transmission of E. coli associated pyometra between two animals. These results could impact the management of sites where several females cohabit in the same local area, such as kennels. Thus, we believe this paper should be publish and it can be of value to vets and kennels.

Reviewer 3 Report

The present manuscript aims to describe the possibility of transversal transmission of pyometra between two partner bitches. The results of the study are interesting and well presented. Since it’s just two individuals, presenting the results as a short communication is the best choice for these results. In fact, I would encourage the authors to go further on this research field.

As already said, the content of the manuscript is interesting, but some changes and modifications are needed.

ABSTRACT

L17. What do D23 and D24 mean? It is explained in the body of the manuscript, but not in the abstract. Just by reading, one can’t guess that they mean the two bitches. On the other hand, why naming them as D23 and D24?

As a general comment on the abstract content, not all the readers of the article will know about genomics. So, maybe, it would be interesting to define the abbreviatures included here.

L66. Ovarian sampling has not been mentioned before. The authors state just uterine and rectal samples. Please, clarify.

Figure 1. Please, add the corresponding letter to each image of this Figure. In addition, the number in the size bars is impossible to identify.

Figure 1 caption. (A) This reviewer is able to say that, actually, there’s blood inside the uterine lumen. However, the image doesn’t allow to clearly observe the presence of neutrophils. The image looks blurry. Same comment goes for glandular content. Identifying neutrophils inside the glandular lumen is totally impossible. (C) Same comment. The content inside the glands can’t be identified by this picture. I would recommend to modify the text and say that B and D are magnifications of A and C respectively and they show the neutrophilic content and so on.

This reviewer is not sure about stating that intestinal E coli induced pyometra. It is not that that particular E coli decided to travel from the intestine to the uterus. The thing is that vaginal microbiota is of faecal origin and is this vaginal microbiota the one that causes pyometra.

L217-219. This sentence seems to state that E coli can be considered as a zoonotic microorganism. I would reconsider this sentence, since E coli is a very ubiquitous bacteria and is physiologically found in the human being as well. In addition, the authors suggest that transmission from bitch to bitch is probably due to faeces and vaginal discharge. Well, human behaviour precludes this infectious pathway for human beings.

Author Response

The present manuscript aims to describe the possibility of transversal transmission of pyometra between two partner bitches. The results of the study are interesting and well presented. Since it’s just two individuals, presenting the results as a short communication is the best choice for these results. In fact, I would encourage the authors to go further on this research field. As already said, the content of the manuscript is interesting, but some changes and modifications are needed.

Authors: Thank you for this comment and for encouraging our group to keep working on this topic.

 L17. What do D23 and D24 mean? It is explained in the body of the manuscript, but not in the abstract. Just by reading, one can’t guess that they mean the two bitches. On the other hand, why naming them as D23 and D24?

Authors: We named the dogs according to the number of samples received during the study (this E. coli transmission was detected during an already published larger experiment (https://doi.org/10.3390/vetsci9050245). The letter D+number was used for each dog, while S+number was used to designated rectal swabs and P+number was used to designate the uterine content for each animal. We agree that it would be better to name these dogs simply 1 or 2, but it would be complicated once these IDs already exist in our library. Also, specifically for this paper, we already made de deposit of the referred genomes with this sample identification (D23 and D24), so it would be confusing for those researchers that want to used our assembled genomes. To make clearer, we added a better explanation of this naming procedure in the manuscript and also in the abstract, as suggested by you.

As a general comment on the abstract content, not all the readers of the article will know about genomics. So, maybe, it would be interesting to define the abbreviatures included here.

Authors: Revised as suggested.

L66. Ovarian sampling has not been mentioned before. The authors state just uterine and rectal samples. Please, clarify.

Authors: We have modified the L55. Samples from the uteri and ovaries were collected for histopathological analysis.

Figure 1. Please, add the corresponding letter to each image of this Figure. In addition, the number in the size bars is impossible to identify.

Authors: The bars were edited and the letters were added as suggested.

Figure 1 caption. (A) This reviewer is able to say that, actually, there’s blood inside the uterine lumen. However, the image doesn’t allow to clearly observe the presence of neutrophils. The image looks blurry. Same comment goes for glandular content. Identifying neutrophils inside the glandular lumen is totally impossible. (C) Same comment. The content inside the glands can’t be identified by this picture. I would recommend to modify the text and say that B and D are magnifications of A and C respectively and they show the neutrophilic content and so on.

Authors: The legend of the figure was revised as suggested.

This reviewer is not sure about stating that intestinal E coli induced pyometra. It is not that that particular E coli decided to travel from the intestine to the uterus. The thing is that vaginal microbiota is of faecal origin and is this vaginal microbiota the one that causes pyometra.

Authors: In the present study, we showed that the same clone of E. coli was part of the intestinal microbiota of one animal and also causing pyometra in two bitches. In fact, it is a consensus that the source of bacteria causing pyometra is indeed the intestinal tract. Like the reviewer stated, the vaginal microbiota reflects the intestinal microbiota, so the common assumption is that the bacteria travel from the intestine to the uterus, passing by the vagina, as stated. So, we really believe it is okay to state that the intestinal E. coli caused the pyometra. It is important to note that, by phenotypic and genomic methods, we showed that the isolates from the intestine from one dog were the same clone from those isolates recovered from the uterus, confirming this hypothesis.

L217-219. This sentence seems to state that E coli can be considered as a zoonotic microorganism. I would reconsider this sentence, since E coli is a very ubiquitous bacteria and is physiologically found in the human being as well. In addition, the authors suggest that transmission from bitch to bitch is probably due to faeces and vaginal discharge. Well, human behaviour precludes this infectious pathway for human beings.

Authors: Revised as suggested.

Round 2

Reviewer 1 Report

Thank you for making some of the changes requested. I still feel there are no adequate alternate explanations for the pyometras  in both dogs. This could still be a coincidence or genetic predisposition. For example, these were sisters - therefore, it is still possible that there was a genetic predisposition to developing CEH. This should be explored a little more in the discussion. Also, were there other dogs in the household? If so, who is to say that the bacteria weren't transmitted from one of the other dogs. What you describe is a likely scenario of the E. coli transmitted from one  dog to the other but alternate explanations should also be discussed. The infection happens because the dogs have uterine pathology, the normal flora from the vaginal canal can ascend into the uterus during estrus and then the cervix closes in diestrus. It seems to me that it is dangerous to imply that a pyometra or even the infection was transmitted because all that happened was that the dogs sniffed each other's butts, have similar gut flora and then both happened to develop a pyometra on the same cycle. 

Author Response

A new paragraph, stating the main limitations raised during the review, was added to the manuscript. Also, the information regarding the presence of other dogs in the same household was also added as suggested. Thanks for pointing these limitations and for your help to improve the quality of this work!